# The robust estimation of examinee ability based on the four-parameter logistic model when guessing and carelessness responses exist

**Xiaozhu Jian**[1,2☺], **Dai Buyun**[ID][3☺]*, **Deng Yuanping**[4☺]

**1** Educational Department, Guangxi Normal University, Guilin, Guangxi Zhuang Autonomous Region, China, **2** School of Educational, Jinggangshan University, Ji'an, Jiangxi Province, China, **3** School of Psychology, Jiangxi Normal University, Nanchang, Jiangxi Province, China, **4** School of Educational, Jimei University, Xiamen, Fujian Province, China

☺ These authors contributed equally to this work.

* 99341655@qq.com

**Data Availability Statement:** All relevant data are within the manuscript and its Supporting Information files.

## Abstract

The three-parameter Logistic model (3PLM) and the four-parameter Logistic model (4PLM) have been proposed to reduce biases in cases of response disturbances, including random guessing and carelessness. However, they could also influence the examinees who do not guess or make careless errors. This paper proposes a new approach to solve this problem, which is a robust estimation based on the 4PLM (4PLM-Robust), involving a critical-probability guessing parameter and a carelessness parameter. This approach is compared with the 2PLM-MLE(two-parameter Logistic model and a maximum likelihood estimator), the 3PLM-MLE, the 4PLM-MLE, the Biweight estimation and the Huber estimation in terms of bias using an example and three simulation studies. The results show that the 4PLM-Robust is an effective method for robust estimation, and its calculation is simpler than the Biweight estimation and the Huber estimation.

## Introduction

One of the major concerns in item response theory (IRT) is the robust estimation of latent ability. Response disturbances such as guessing, cheating, carelessness, and transcription errors often cause biases in estimated latent ability, are determined by a maximum likelihood estimator (MLE). Waller (1974), with Wainer and Wright (1980), proposed approaches to the robust estimation of latent ability [1, 2], including traditional correction for guessing, Jackknife, AMT-Robustified Jackknife, and WIM. AMT-Robustified Jackknife was generally considered the best and most efficient method for tests comprising forty or fewer items [2]. Mislevy and Bock (1982) proposed the Biweight estimator, a robust-estimation method using the MLE, and compared it with the two-parameter Logistic model(2PLM). The results showed that the Biweight estimator could typically reduce biases, thereby eliminating measurement disturbances [3]. Schuster and Yuan (2011) summarized two strategies to robustify ability estimation in the presence of disturbances. The first strategy was trimming inconsistent item response,

**Funding:** The study was supported by Social Science Foundation of Jiangxi Province (19JY02, 17JY16) received by XJ.

**Competing interests:** The authors have declared that no competing interests exist.

which was not considered the best method [4]. Waller (1974) assumed that the data produced were not consistent with the expected probability according to subjects' ability and attempted to trim inconsistent or uninformative responses from each subject's vector of responses [1]. The second strategy was giving less weight to observation responses which are prone to response disturbance. This type of strategy includes Biweight estimation and Huber estimation. Schuster and Yuan (2011) proposed the Huber estimation to robustify the ability estimation [4]. According to their investigation, the Huber estimation, rather than the Biweight estimation, should be applied when reducing sampling variability was prior to reducing biases. But these strategies still have some faults and needs to be improved. So in the study, a robust estimation based on the 4PLM (4PLM-Robust) is proposed to robustify the ability estimation.

## The IRT model and four-parameter logistic model

As we know, the IRT models include the Rasch model and the two-parameter Logistic model (2PLM) [5]. The 2PLM is described as

$$
\begin{aligned}
P_{vi} &= \frac{\exp(a_i(\theta_v - b_i))}{1 + \exp(a_i(\theta_v - b_i))}, \\
&= \psi(a_i(\theta_v - b_i))
\end{aligned}
\tag{1}
$$

where $\theta_v$ is the ability of subject $v$, $a_i$ is the discrimination parameter of item $i$, and $b_i$ is the difficulty parameter of item $i$.

Concerning guessing responses in tests, Birnbaum (1957) proposed a three-parameter Logistic model (3PLM) [5], which was expressed as

$$
P_{vi} = c_i + (1 - c_i)\frac{\exp(a_i(\theta_v - b_i))}{1 + \exp(a_i(\theta_v - b_i))}.
\tag{2}
$$

Barton and Lord (1981) explored whether changing the upper asymptote could improve standardized test scoring [6]. Barton and Lord (1981) therefore influenced the upper-asymptote model by adding a fourth parameter, $\gamma$, dropping the upper asymptote to below 1:

$$
P_{vi} = c_i + (\gamma_i - c_i)\frac{\exp(a_i(\theta_v - b_i))}{1 + \exp(a_i(\theta_v - b_i))}
\tag{3}
$$

where $\gamma_i$ denotes the upper asymptote parameter of item $i$. When $\gamma_i = 1$ and $c_i = 0$, the 4PLM becomes the 2PLM; i.e., the 2PLM is a special case of the 4PLM.

Some researcher use the $\gamma$ parameter to denote the upper asymptote [7]. Barton and Lord (1981) re-estimated the test scores of thousands of students taking four separate exams to determine the effects of fixing $\gamma$ at 0.99 and 0.98 and concluded that the changes were too small to be of practical significance [6].

In this article, the 4PLM-based robustified approach, abbreviated as 4PLM-Robust, serves the purpose of robust estimation in the presence of response disturbances including random guessing and carelessness.

**Recent literature on the 4PLM.** Waller and Reise (2009) reviewed the literature on the 4PLM up to 2009 [8]. Barton and Lord (1981) presented negative opinions on the 4PLM, echoed by Hambleton and Swaminathan (1985). Perhaps, their comments influence the research. The research on the 4PLM stagnated for a quarter of a century.

Still, Reise and Waller (2003) suggested that the 4PLM may be uniquely appropriate for characterizing psychopathology data in non-adaptive tests [9], and their study had been credited with reviving interest in the 4PLM. The simulation study by Jian, Dai, and Peng (2007) showed that occasional errors made by relatively high-ability subjects on easy items resulted in

relatively larger biases on non-adaptive tests or paper-and-pencil tests. However, these biases may be effectively reduced using the 4PLM [10].

In the computerized adaptive testing, Rulison and Loken (2009) argued that it was significant that the upper asymptote's being <1 allowed a small probability of error even by extremely high-ability students [11]. In their findings, the bias and root mean square error (RMSE) for high-ability students could be lowered using the four-parameter Logistic model and a less informative prior with the first two items missing because the computerized adaptive testing (CAT) algorithm ascended more quickly after initial underperformance. Liao, Ho, Yen, and Cheng (2012) observed that these results are similar to Rulison and Loken's (2009) and indicated that the 4PLM provided a more robust ability estimation than the 3PLM in CAT [12]. However, Green (2011) expounded his views that there are rare errors on middle difficulty items for very proficient test-takers, and the so-called bias was one of selective data analysis [13]. The authors believe that whether the score penalty for very proficient test-takers is statistically significant, the faultiness of underestimating them in the Rasch model or 2PLM in the test has been found. Moreover, the reasons and methods should also be found in different research angles.

Loken and Rulison (2010) employed a Bayesian approach to successfully recover parameter estimates for items and respondents [14]. Their study provided specified for $c$ and 03 $\gamma$ in the 4PLM model. As implemented in BRUGS, the Gibbs sampler, an open-source R package based on OpenBUGS architecture (Thomas, 2006) [8], was used by Waller and Reise (2009) to estimate the 4PLM. Furthermore, the WINSETPS program (Linacre, 2013a, 2013b) was available to estimate the upper asymptote in the 4PLM, and the upper asymptote can be observed as the approximation of the $\gamma$ parameter [15, 16].

**The method of the 4PLM-robust.** A new approach, a robust estimation based on the 4PLM (4PLM-Robust) is proposed.

At first, the maximum-likelihood estimation of ability is illustrated. Let $u_i$ denote the response of an individual to the $i$-th item. The probability of a correct response is denoted by $P(\eta_i)$, where $\eta_i$ is an expression involving ability and item parameters in 2PLM. The probability of a single response $u_i$ is given by

$$P(\eta_i)^{u_i} \cdot (1 - P(\eta_i))^{(1-u_i)}$$

The logarithm of this probability, $l_i = u_i \cdot \log(P(\eta_i)) + (1-u_i) \cdot \log((1-P(\eta_i)))$, is the log likelihood contribution to the item response. If local independence holds, these contributions can be added. Taking derivatives of ability and setting the result at zero yields

$$\sum_i \frac{u_i - P(\eta_i)}{P(\eta_i) \cdot (1 - P(\eta_i))} \cdot \frac{dP(\eta_i)}{d\theta} = 0 \tag{4}$$

Using the Logistic model, $P(\eta_i) = \exp(\eta_i)/(1+\exp(\eta_i))$, and noting that $\frac{dP(\eta_i)}{d\theta} = \frac{dP(\eta_i)}{d\eta_i} \cdot \frac{d\eta_i}{d\theta}$, so Eq 4 simplifies to

$$\sum_i (u_i - P(\eta_i)) \cdot \frac{d\eta_i}{d\theta} = 0 \tag{5}$$

Let $\eta_i = r_i = a_i(\theta - b_i)$, and $P(\eta_i) = \exp(r_i)/(1+\exp(r_i))$
Using the 3PLM, that is, $P(\eta_i) = \exp(r_i)/(1+\exp(r_i))$, Eq 5 can be transformed into

$$\sum_i a_i \cdot (u_i - P(\eta_i)) = 0 \tag{6}$$

Using the 3PLM, that is, $P(\eta_i) = c_i + (1 - c_i) \cdot \frac{\exp(r_i)}{(1+\exp(r_i))}$, Eq 5 can be transformed into

$$\sum_i a_i \cdot (u_i - P(\eta_i)) \cdot (P(\eta_i) - c_i)/[P(\eta_i)(1 - c_i)] = 0 \tag{7}$$

If $c_i = 0$, Eq 7 will be the same as Eq 6.

Using the 4PLM, that is, $P(\eta_i) = c_i + (\gamma_i - c_i) \cdot \frac{\exp(r_i)}{(1+\exp(r_i))}$, Eq 5 can be transformed into

$$\sum_{i=1} a_i \cdot (u_i - P(\eta_i)) \cdot (\gamma_i - P(\eta_i)) \cdot \frac{(P(\eta_i) - c_i)}{P(\eta_i) \cdot (1 - P(\eta_i)) \cdot (\gamma_i - c_i)} = 0 \tag{8}$$

If $c_i = 0$ and $\gamma_i = 1$, Eq 8 will be the same as Eq 6.

In this article, a robust estimation based on the 4PLM, abbreviated as 4PLM-Robust, is proposed, and it is an alternative approach to robustifying ability estimation. The 4PLM-Robust estimation formulated as follows:

$$\begin{cases} c_i = \begin{cases} 0.20 & for\ P(r_i) \leq p^* \\ \\ 0 & \text{otherwise} \end{cases} \\ \gamma_i = 1 - p^{**} & \text{carelessness errors appear} \end{cases} \tag{9}$$

where $0 \leq p^* < 0.20$ and $0 \leq p^{**} \leq 0.03$.

If the guessing and carelessness response disturbances do not occur simultaneously on the $i$-th item, then $p^* = 0$ and $p^{**} = 0$. If $p^* = 0$ and $p^{**} = 0$, then $c_i = 0$ and $\gamma_i = 1$ in Eq 8. If $c_i = 0$ and $\gamma_i = 1$, 4PLM-Robust estimation become 4PLM. That is, 4PLM-Robust estimation originate in 2PLM and can be simplified as 2PLM. In other words, 2PLM is one special case of 4PLM-Robust estimation.

The subjects only guess when item difficulty extends beyond their abilities to the extent where their knowledge cannot bring them apparent benefit, and in such cases, the subjects most likely choose randomly from among the four or five alternative responses. As demonstrated by Mislevy and Bock (1982), subjects are assumed to have guessed on the item if their expected probability of correct responding [3], $P(r_i) = \exp(r_i)/(1+\exp(r_i))$, is below the $p^*$ value at 0.05, or 0.10, or 0.15, because the probability of a random choice from five alternatives is equal to 0.20. If the expected probability $P(\theta)$ is lower than the critical point $p^*$, the subject is apt to respond by random guessing. Thus, $p^*$ denotes a critical point for the probability of random guessing, and is noted as a critical-probability guessing parameter. In the paper, $\psi(\theta_v - b_i)$ represents the correct response probability of the subject $v$ using the 2PLM. For example, when $p^* = 0.10$ is deemed as the average probability for guessing on the $i$-th item, $\psi(\theta_v - b_i) \leq 0.10$ indicates $r_i = a_i(\theta - b_i) \leq -2.20$. $\psi(\theta_v - b_i) < p^*$ indicates $\psi(\theta_v - b_i) < 0.05$ and $r_i = a_i(\theta - b_i) \leq -1.73$ when $p^*$ is at 0.05, and $\psi(\theta_v - b_i) < 0.15$ and $r_i = a_i(\theta - b_i) \leq -2.97$ when $p^* = 0.15$. Moreover, a setting of $p^* = 0.15$ signals a high level of guessing in a test, and $p^* = 0.05$ indicates low guessing in a test, and $p^* = 0.00$ indicates no guessing.

Mislevy and Bock (1982) also proposed the carelessness parameter $p^{**}$, signifying the probability of occasional error because of carelessness [3]. Schuster and Yuan (2011) expounded $p^{**}$ as an occasional error probability, which reflected the result of carelessness, transcription errors, test anxiety and exorbitant intention [4]. Furthermore, they noted that error probability $p^{**}$ was not dependent on subject ability or item difficulty. In the present study, given that carelessness is a small probability event in school achievement testing, it is believed that $p^{**}$ is a small probability, i.e., $0 \leq p^{**} \leq 0.03$. Similarly, the approach of the 4PLM-Robust to robustify ability estimation in the presence of careless responses can be expressed as $\gamma_i = 1 - p^{**}$.

However, defining or identifying the occasional error probability in practice tests is complicated. If $p^{**} = 0.01$, it can be assumed that one item may be missed on a one hundred-item test by an occasional error, so $p^{**} = 0.01$ is a low probability of occasional error. If $p^{**} = 0.03$, it can be assumed that one item may be missed on a thirty-three-item test by an occasional error, so $p^{**} = 0.03$ is a high probability of occasional error.

In the Eq 8, if $p^* = 0.15$ and $p^{**} = 0.03$, the 4PLM-Robust is abbreviated to the 4PLM-Robust ($p^* = 0.15$, $p^{**} = 0.03$).

## The Biweight and Huber robust estimation

In the Biweight, Huber robust estimation, two decisions determine the weight attached to each item: (a) the definition of the residual between estimated ability and item difficulty, $r_i = a_i(\theta - b_i)$; (b) the selection of the weight function $w(r_i)$.

In the Biweight, and Huber robust estimation, the robust estimate of ability requires a modified likelihood equation solution, that is, Eq 5 is weighted by $w(r_i)$. So Eq 5 becomes as:

$$\sum_{i=1} w(r_i) \cdot (u_i - P(\eta_i)) \cdot \frac{dr_i}{d\theta} = 0 \tag{10}$$

In Biweight estimations, the weight function $w(r_i)$ is shown as

$$w(r_i) = \begin{cases} [1 - (r_i/B)^2]^2 & For \ |r_i| \leq B \\ 0 & For \ |r_i| > B \end{cases} \tag{11}$$

The so-called tuning constant $B$ determines the degree to which residuals are downweighted. Large values of $B$ mean little downweighting and small values mean considerable downweighting. Mislevy & Bock (1982) used B = 4.0. Thus, as $r_i$ exceeds 4.0, the weight becomes zero, eliminating the observation from the estimating. If $r_i = 0$, then $w(r_i) = 1$ in Eq 10 and Eq 9. And then Eq 9 will become the same as Eq 5. That is, 2PLM is the special case of Biweight estimation.

In Huber estimations, the weight function $w(r_i)$ is shown as

$$w(r_i) = \begin{cases} 1 & For \ |r_i| \leq H \\ H/|r_i| & For \ |r_i| > H \end{cases} \tag{12}$$

In the Huber estimation, Schuster and Yuan (2011) used $H = 1.0$ [4]. The weight function $H$ does not downweight responses at all as long as $|r_i| \leq H$, that is, $w(r_i) = 1$ in Eq 11 and in Eq 9, and then Eq 9 becomes the same as Eq 5. That is, 2PLM is the special case of Huber estimation.

Therefore, Biweight and Huber estimations do downweight the responses by the weight function $w(r_i)$ and the tuning constants $B$ or $H$. By comparison, the 4PLM-Robust method robustify ability estimation by c, $\gamma$ parameter in the 4PLM according to $r_i = a_i(\theta - b_i)$, so the 4PLM-Robust estimations do not downweight the responses.

**Present study.** The generated model about the carelessness error in the Schuster and Yuan (2011), $(1-\gamma)P_j + \gamma Q_j$, is not exactly. For instance, for the low-ability examinee, if the response probability $P_j = 0.40$ and $\gamma = 0.05$, the probability $P'$ will be $P' = (1-0.05) \times 0.40 +- 0.05 \times 0.60 = 0.41$, that is, $P' > P_j$. As it is known, $P' > P_j$ is paradoxical and illogical. Moreover, the generated model mentioned above just only includes the high-ability examinees.

In this article, the authors believe that the generated model about the carelessness error is proposed according to the Principle About Guessing and Carelessness Error, and the Robust estimation should also be proposed according to it.

This paper comprises two sections. First, a ten-item test is applied to exemplify the comparisons among the six methods for robustifying ability estimates, which include the maximum likelihood estimation using the two-parameter Logistic model (2PLM-MLE), the maximum likelihood estimation using the 3PLM (3PLM-MLE), the maximum likelihood estimation using the 4PLM (4PLM-MLE), the Biweight estimation (BIW), the Huber weight estimation (Huber), and the maximum likelihood estimation using the 4PLM-Robust ($p^* = 0.15$, $p^{**} = 0.02$). Second, three simulation studies are presented, respectively in three test cases to comprehensively compare the six estimation approaches comprehensively.

## Example

Assume that the difficulty parameters of a 10-item test are the values $b_1 = -4$, $b_2 = -3$, $b_3 = -2$, $b_4 = -1$, $b_5 = 0$, $b_6 = 0$, $b_7 = 1$, $b_8 = 2$, $b_9 = 3$, $b_{10} = 4$; and the discrimination parameters of the first five items are 1.0 and those of the last five items are 0.8. Besides, 11 fictitious subjects respond to the test. Subject #1 answers the 5 easy items correctly and misses the 5 difficult items; thus, the response pattern of subject #1 is designated as the referenced pattern. His estimated ability is 0.061 using the 2PLM-MLE, which is perceived as the referenced ability value. Subjects #2-#6, respectively answer 6 items correctly. Subject #6 correctly responds to the hardest item, i.e., the $10^{th}$ item. Subjects more often respond correctly to easy items but less often answer correctly to hard ones [3]; thus it is reasonable to believe that subject #6 applies effort to guess the $10^{th}$ item. The aberrant response of subject #6 is thus regarded as measure disturbance. Subjects #7-#11 answer four items correctly, respectively. Subject #11 misses the easiest item, which is also viewed as an aberrant response, or measure disturbance.

Table 1 shows the six estimators for the 11 subjects. The ability estimates in columns 3 through 5 are obtained by BILOG [17]. The ability estimates in columns 6 through 8 are obtained using the Visual Basic program because of the lack of corresponding programs in BILOG.

## Method 1: Maximum likelihood estimation using the 2plm(2PLM-MLE)

The maximum likelihood using the 2PLM estimates is identical for subject #2 to #6, and so is for subject #7 to #11. Thus, in this case, the different response patterns cannot be differentiated

**Table 1. The biases of the ability estimates of the five approaches.**

| Subject # | Responses | BILOG / Visual Basic Program | | | Visual Basic Program | | |
|---|---|---|---|---|---|---|---|
| | | 2PLM | BIW | 3PLM | 4PLM | 4PLM | Huber |
| | | -MLE | B = 4 | -MLE | -MLE | -Robust | H = 1 |
| 1 | 1111100000 | 0.061 | 0.048 | -0.221 | -0.133 | 0.068 | 0.052 |
| 2 | 1111110000 | 0.763 | 0.710 | 0.538 | 0.627 | 0.724 | 0.711 |
| 3 | 1111101000 | 0.763 | 0.746 | 0.436 | 0.494 | 0.762 | 0.711 |
| 4 | 1111100100 | 0.763 | 0.653 | 0.262 | 0.131 | 0.206 | 0.666 |
| 5 | 1111100010 | 0.763 | 0.421 | 0.061 | -0.060 | 0.096 | 0.346 |
| 6 | 1111100001 | 0.763 | 0.163 | -0.080 | -0.114 | 0.073 | 0.257 |
| 7 | 1111000000 | -0.755 | -0.726 | -0.999 | -0.886 | -0.709 | -0.720 |
| 8 | 1110100000 | -0.755 | -0.798 | -1.239 | -1.352 | -0.684 | -0.720 |
| 9 | 1101100000 | -0.755 | -0.617 | -1.437 | -1.817 | -0.479 | -0.435 |
| 10 | 1011100000 | -0.755 | -0.160 | -1.570 | -0.535 | -0.022 | -0.224 |
| 11 | 0111100000 | -0.755 | 0.048 | -1.642 | -0.189 | 0.056 | -0.146 |

*Note*. 3PLM-MLE indicates c = 0.20 for items 1~10 using the 3PLM.

4PLM-MLE indicates c = 0.20 for items 1~10, and $\gamma$ = 0.95 for items 1~10 using the 4PLM.

4PLM-Robust indicates 4PLM-Robust ($p^* = 0.15$, $p^{**} = 0.02$).

by the 2PLM-MLE. Similar results have also been observed by other researchers (Embretson & Reise, 2000) [18].

The item difficulty of the 10-*th* item ($b_{10} = 4$) is greatly larger than the ability of Subjects #6, but Subject #6 gives a correct response on the 10-th item by guessing probably and get the same ability value as Subject #2. So there is a guessing response probably, and the ability estimation of Subject #6 is not robust.

The item difficulty of the1-*st* item ($b_1 = -4$) is greatly smaller than the ability of Subject #11, but he or she gives a wrong response on the 10-th item because of carelessness or other error probably and gets the same ability value as Subject #7. So there is a carelessness response probably, and the ability estimation of Subject #11 is not robust.

## Method 2: Using the Biweight estimation (BIW estimation)

The BIW estimates for subject #2 to #6 differ from one another, as do the BIW estimates for subject #7 to #11. The score on the 10-*th* item of Subject #6 can be seen as most seriously deviated from the referenced pattern of subject #1, because on the probability it can be conclude that Subject #6 has a guessing response on the 10-*th* item. so the BIW estimate disregards his unexpectedly correct response to item 10 and sets the ability estimate to 0.163; i.e., the influence of the guessing response in subject #6 is reduced. Moreover, the responses from Subject #5 and Subject #4 deviate from the referenced pattern, and the BIW estimate assigns them ability estimates of 0.421 and 0.653, respectively. The estimated abilities of subjects #2 and #3, according to the BIW, are close to the estimated abilities using the 2PLM-MLE.

Conversely, subject #11 missed the easiest item, and his response pattern deviates from the referenced pattern of subject #1. Accordingly, the BIW estimate largely disregards his unexpectedly correct response to item 1 and sets the ability estimate to 0.048, which reduces the influence of the carelessness responses of subject #11. The responses of subject #10 and subject #9 also deviate from the referenced pattern. The BIW estimate assigns them ability estimates of -0.160 and -0.617, respectively. The estimated abilities of subjects #7 and #8, using the BIW, are close to their estimated abilities using the 2PLM-MLE.

## Method 3: Maximum likelihood estimation using the 3PLM (3PLM-MLE)

First, *c* is assigned as 0.2 from item 1 to 10. Using the 3PLM-MLE, the estimate for subject #6 is -0.080, which is quite close to the estimate for subject #1 compared with subject #2 to #5. The trend of the estimates for subject #2 to #6, using the 3PLM-MLE, is similar to that of the estimates using the BIW.

All of the subjects' estimates using the 3PLM-MLE are lower than those which use the 2PLM-MLE. Notably, the estimates for subject #8 to #11 are all underestimated by a large margin.

## Method 4: Maximum likelihood estimation using the 4PLM (4PLM-MLE)

In item 1 to 10, *c* is assigned as 0.2, and *γ* is assigned as 0.95. The estimates for subject #2 to #6, using the 4PLM-MLE, are close to the estimates using the 3PLM-MLE. The estimates of subject #5 and #6 are close to the estimate of subject #1, using the 4PLM-MLE; i.e., the influences of the guessing responses of subject #5 and #6 have been decreased.

Using the 4PLM-MLE and the BIW, the ascending trends of subject #8 to #11 are similar, and the estimate of subject #11 is close to that of subject #1. The influences of the guessing responses of subject #6 and the carelessness responses of subject #11 are decreased. However, the estimates of subject #8 and #9, using the 4PLM-MLE, are underestimated compared with those using the 2PLM-MLE and the 3PLM-MLE.

## Method 5: Maximum likelihood estimation using the 4PLM-Robust ($p^* = 0.10$, $p^{**} = 0.02$)

If $p^* = 0.10$ and $p^{**} = 0.02$ in Eq 5, the 4PLM-Robust is abbreviated to the 4PLM-Robust ($p^* = 0.10$, $p^{**} = 0.02$) in the paper.

The trend of the estimates for subject #2 to #6 using the 4PLM-Robust is similar to the trend derived using the BIW, the 3PLM-MLE and the 4PLM-MLE. In addition, the trend of the estimates for subject #7 to #11 using the 4PLM-Robust is similar to the trend using the BIW. The estimates of subject #5 and #6 are close to the estimate of subject #1 using the 4PLM-Robust; i.e., the influences of the guessing responses of subject #5 and #6 can effectively be decreased.

The ascending trend of the 4PLM-Robust for subject #7 to #11 is identical to that of the BIW. The estimates of subject #10 and subject #11 are close to the estimate of subject #1; i.e., the influences of the carelessness responses on subject #10 and subject #11 can be decreased effectively.

## Method 6: The Huber estimation

The decreasing trend of the Huber estimates for subject #2 to #6 is identical to that of the BIW estimates, although there are differences between them. The ascending trends of the Huber estimates and the BIW estimates for subject #7 to #11 are also identical. As Schuster and Yuan (2011) illustrated, Huber can robustify ability estimation for aberrant responses.

In summary, the 'Example' can show the guessing and carelessness responses, and the authors try to compare the several robust estimation methods.

## Simulation study

This section compares the six approaches when different types of disturbing responses exist. The hypothetical test instrument contains 45 items, with difficulty thresholds ranging from -4.4 to 4.4 in steps of 0.2. And log (a) ~ $N(0,1)$. Using the 3PLM-MLE and the 4PLM-MLE, $c$ parameters of the items obey $N(0.2, 0.01)$, and $\gamma$ parameters of all items are set to 0.97. Thus, item parameters using the 2PL-MLE, the 3PL-MLE, and the 4PL-MLE are all known in simulation cases; ability parameters need only to be estimated. The response vectors for the five subjects are generated with abilities of -2.00, -1.00, 0.00, 1.00, and 2.00, with response vectors for these subjects generated by 5,000 replications. The two-parameter Logistic model is the true generated-response model in three simulation cases with two types of response disturbances, i.e., random guessing and carelessness. The first case includes response disturbances by random guessing without carelessness, the second case includes response disturbances by carelessness without random guessing, and the third case includes response disturbances by simultaneous random guessing and carelessness.

### Simulation Case I: Random guessing without carelessness

The first type of disturbance is considered random guessing. Responses indicating random guessing were generated by the following model (Mislevy & Bock, 1982) [3]:

$$\mathrm{P}_{vi} = \begin{cases} 0.20 & \text{if } \psi(\theta_v - b_i) \leq p* \\ \psi(\theta_v - b_i) & \text{otherwise} \end{cases}, \tag{13}$$

where $\psi(\theta_v - b_i)$ represents the correct response probability of the subject $v$ using the 2PLM. The generated model is essentially the 2PLM plus the formula of guessing disturbance. Four different values for $p^*$ are set to 0.00, 0.05, 0.10, and 0.15.

In Simulation Case I there is no carelessness; the approach using the 4PLM-Robust to robustify ability estimation is as follows:

$$
\begin{cases}
c = \begin{cases} 0.20 & if \;\; \psi(\theta_v - b_i) < p* \\ 0 & \text{otherwise} \end{cases} \\
\gamma = 1 & \text{for all items without carelessness errors}
\end{cases}
, \tag{14}
$$

where the critical-probability guessing parameter $p^*$ can be set as 0.00, 0.05, 0.10, or 0.15 according to guessing on the test.

Table 2 shows the biases between the estimated ability and the true ability value in Simulation Case I, varying combinations of the true ability value and $p^*$. In Table 2, the BIW (B = 4) represents the BIW estimation with the BIW iterations from maximum likelihood estimates [3], with B = 4. Huber (H = 1) represents the Huber estimation with Huber iterations from maximum likelihood estimates, with H = 1 [4].

The first section of Table 2 shows the results when the response-generated model is set at $p^*$ = 0.00, which indicates no guessing. The absolute values of the biases of the 2PLM-MLE, the BIW estimation, the Huber estimation, and the 4PLM-Robust ($p^*$ = 0.00) of these subjects are minimal, less than 0.02. The absolute values of the biases of the 4PLM-Robust ($p^*$ = 0.05), the 4PLM-Robust ($p^*$ = 0.10), and the 4PLM-Robust ($p^*$ = 0.15) of the five subjects become successively larger. Because responses from guessing are generated with $p^*$ = 0.00, which indicates no guessing disturbance, the absolute values of the 4PLM-Robust ($p^*$ = 0.00) biases are small, close to zero. However, the absolute values of the 4PLM-Robust biases with $p^*$ = 0.05, 0.10, and 0.15 are slightly larger. Moreover, the biases, using the 3PLM-MLE and the 4PLM-MLE, are all negative values, and their absolute values are relatively large, as shown in the other sections of Table 2. This result indicates that the 3PLM-MLE and the 4PLM-MLE severely underestimate the subjects.

The second section of Table 2 shows the results when the response-generated model is set at $p^*$ = 0.05. The biases of the 2PLM-MLE, the Huber estimation, and the 4PLM-Robust ($p^*$ = 0.00) indicate that those biases of the lower-ability subjects are larger than those of the higher-ability subjects. Compared with the results when $p^*$ = 0.00, the bias of the 2PLM-MLE is 0.722 at a true ability level of -2.00, as is the other lower-ability subject. Conversely, the absolute values of the biases of the 4PLM-Robust ($p^*$ = 0.05) are not larger than 0.01. The absolute values of the biases of the 4PLM-Robust ($p^*$ = 0.10) and the 4PLM-Robust ($p^*$ = 0.15) also decreased, compared with the biases of the 2PLM-MLE. The biases which use the 3PLM-MLE and the 4PLM-MLE are all negative values, and their absolute values are relatively large.

The third section of Table 2 shows the results when the response-generated model is set at $p^*$ = 0.10. Along with the increase of $p^*$ value, the absolute values of the biases of the 2PLM-MLE, the BIW estimation, the Huber estimation, and the 4PLM-Robust ($p^*$ = 0.00) increase compared with the result at $p^*$ = 0.05 and $p^*$ = 0.00. Simultaneously, the absolute values of the biases of the 4PLM-Robust ($p^*$ = 0.10) are nearly zero; in other words, measure disturbances by guessing are reduced. This is also visible in the fourth section of Table 2, in which the biases of the 4PLM-Robust ($p^*$ = 0.15) are nearly zero.

Consequently, if the critical point $p^*$ of the 4PLM-Robust is equal to that of the guessing responses in the generated model, the biases of the 4PLM-Robust ($p^*$ = *critical point*) will be nearly zero. The 4PLM-Robust ($p^*$ = *critical point*) is superior to the BIW estimation and the Huber estimation if the guessing disturbance's probability is equal to or greater than 0.05.

**Table 2. The biases in Case I: Random guessing without carelessness.**

| Generated model | Method | True ability value | | | | |
|---|---|---|---|---|---|---|
| | | **-2.000** | **-1.000** | **0.000** | **1.000** | **2.000** |
| $p^* = 0.00$ | 2PLM-MLE | 0.003 | -0.009 | -0.005 | -0.003 | 0.002 |
| | 3PLM-MLE | -0.628 | -0.573 | -0.558 | -0.524 | -0.486 |
| | 4PLM-MLE | -0.552 | -0.495 | -0.461 | -0.415 | -0.354 |
| | 4PLM-Robust ($p^* = 0.00$) | -0.001 | -0.006 | 0.004 | 0.011 | 0.003 |
| | 4PLM-Robust ($p^* = 0.05$) | -0.047 | -0.044 | -0.025 | -0.029 | 0.014 |
| | 4PLM-Robust ($p^* = 0.10$) | -0.078 | -0.063 | -0.072 | -0.071 | -0.050 |
| | 4PLM-Robust ($p^* = 0.15$) | -0.123 | -0.116 | -0.097 | -0.102 | -0.099 |
| | BIW (B = 4) | 0.000 | -0.010 | -0.010 | 0.000 | 0.010 |
| | Huber (H = 1) | -0.010 | 0.000 | 0.000 | 0.000 | 0.010 |
| | 4PLM-Robust ($p^* = 0.10, p^{**} = 0.02$) | -0.038 | -0.020 | -0.006 | 0.024 | 0.018 |
| $p^* = 0.05$ | 2PLM-MLE | 0.722 | 0.483 | 0.289 | 0.132 | 0.021 |
| | 3PLM-MLE | -0.579 | -0.531 | -0.509 | -0.510 | -0.491 |
| | 4PLM-MLE | -0.504 | -0.439 | -0.417 | -0.383 | -0.354 |
| | 4PLM-Robust ($p^* = 0.00$) | 0.710 | 0.521 | 0.287 | 0.098 | 0.042 |
| | 4PLM-Robust ($p^* = 0.05$) | -0.007 | 0.005 | 0.010 | -0.009 | -0.008 |
| | 4PLM-Robust ($p^* = 0.10$) | -0.051 | -0.048 | -0.012 | -0.017 | -0.015 |
| | 4PLM-Robust ($p^* = 0.15$) | -0.111 | -0.096 | -0.087 | -0.082 | -0.071 |
| | BIW (B = 4) | 0.050 | 0.042 | 0.058 | 0.039 | 0.047 |
| | Huber (H = 1) | 0.180 | 0.170 | 0.130 | 0.050 | 0.022 |
| | 4PLM-Robust ($p^* = 0.10, p^{**} = 0.02$) | 0.044 | 0.046 | 0.050 | 0.041 | 0.052 |
| $p^* = 0.10$ | 2PLM-MLE | 0.777 | 0.580 | 0.386 | 0.180 | 0.063 |
| | 3PLM-MLE | -0.535 | -0.500 | -0.495 | -0.473 | -0.476 |
| | 4PLM-MLE | -0.448 | -0.406 | -0.383 | -0.350 | -0.340 |
| | 4PLM-Robust ($p^* = 0.00$) | 0.840 | 0.561 | 0.437 | 0.192 | 0.077 |
| | 4PLM-Robust ($p^* = 0.05$) | 0.165 | 0.161 | 0.142 | 0.080 | 0.032 |
| | 4PLM-Robust ($p^* = 0.10$) | -0.004 | 0.003 | -0.009 | -0.011 | 0.021 |
| | 4PLM-Robust ($p^* = 0.15$) | -0.088 | -0.059 | -0.089 | -0.074 | -0.052 |
| | BIW (B = 4) | 0.102 | 0.096 | 0.095 | 0.078 | 0.044 |
| | Huber (H = 1) | 0.287 | 0.236 | 0.186 | 0.121 | 0.042 |
| | 4PLM-Robust ($p^* = 0.10, p^{**} = 0.02$) | 0.096 | 0.108 | 0.119 | 0.105 | 0.101 |
| $p^* = 0.15$ | 2PLM-MLE | 0.832 | 0.623 | 0.471 | 0.266 | 0.083 |
| | 3PLM-MLE | -0.530 | -0.487 | -0.463 | -0.472 | -0.453 |
| | 4PLM-MLE | -0.436 | -0.375 | -0.354 | -0.344 | -0.318 |
| | 4PLM-Robust ($p^* = 0.00$) | 0.811 | 0.643 | 0.458 | 0.232 | 0.077 |
| | 4PLM-Robust ($p^* = 0.05$) | 0.228 | 0.290 | 0.236 | 0.120 | 0.066 |
| | 4PLM-Robust ($p^* = 0.10$) | 0.063 | 0.065 | 0.041 | -0.023 | -0.032 |
| | 4PLM-Robust ($p^* = 0.15$) | -0.031 | -0.042 | 0.012 | -0.039 | -0.027 |
| | BIW (B = 4) | 0.173 | 0.143 | 0.122 | 0.127 | 0.106 |
| | Huber (H = 1) | 0.311 | 0.270 | 0.198 | 0.143 | 0.153 |
| | 4PLM-Robust ($p^* = 0.10, p^{**} = 0.02$) | 0.131 | 0.146 | 0.130 | 0.147 | 0.145 |

Regarding each examinee's absolute values of biases in Table 2, 4PLM-Robust ($p^* = 0.10$, $p^{**} = 0.02$) is a little larger than the Huber and BIW estimation when $p^* = 0.00$. However,4PLM-Robust ($p^* = 0.10$, $p^{**} = 0.02$) is less than the Huber and the BIW estimation when $p^* = 0.10$ and $p^* = 0.15$.

## Simulation Case II: Carelessness without random guessing

The second type of response disturbance is carelessness. The authors of this paper propose a new generating model for the carelessness disturbance:

$$P_{vi} = \psi(\theta_v - b_i) - \psi(\theta_v - b_i) \bullet p** = \psi(\theta_v - b_i) \bullet (1 - p**), \tag{15}$$

where $\psi(\theta_v - b_i)$ is the expected probability for subject $v$ to answer item $i$ correctly using the 2PLM. The Eq (14) for the generated model is essentially the 2PLM plus the formula of carelessness disturbance. The probability of occasional error is denoted by $p^{**}$. In the 4PLM-Robust, $\gamma = 1-p^{**}$. Three critical point values for $p^{**}$ are set at 0.00, 0.01, and 0.03.

In Case II, the approach using the 4PLM-Robust to robustify ability estimation is perceived as

$$\begin{cases} c = 0 & \text{for all items without random guessing} \\ \gamma = 1 - p** & \text{carelessness errors appear} \end{cases}, \tag{16}$$

where the carelessness parameter $p^{**}$ in the 4PLM-Robust is set at 0.00, 0.01, and 0.03. Table 3

**Table 3. The biases in Case II: Carelessness without random guessing.**

| Generated model | Method | True ability value | | | | |
|---|---|---|---|---|---|---|
| | | -2.000 | -1.000 | 0.000 | 1.000 | 2.000 |
| $p^{**} = 0.00$ | 2PLM-MLE | -0.005 | 0.004 | -0.006 | 0.015 | 0.006 |
| | 3PLM-MLE | -0.629 | -0.571 | -0.557 | -0.513 | -0.485 |
| | 4PLM-MLE | -0.534 | -0.489 | -0.445 | -0.404 | -0.363 |
| | 4PLM-Robust($p^{**} = 0.00$) | 0.005 | -0.009 | -0.001 | -0.006 | 0.003 |
| | 4PLM-Robust($p^{**} = 0.01$) | 0.052 | 0.088 | 0.103 | 0.109 | 0.112 |
| | 4PLM-Robust($p^{**} = 0.03$) | 0.116 | 0.129 | 0.143 | 0.151 | 0.159 |
| | BIW (B = 4) | -0.011 | 0.008 | 0.017 | 0.002 | 0.010 |
| | Huber (H = 1) | -0.007 | 0.005 | -0.008 | -0.004 | -0.012 |
| | PLM-Robust ($p^* = 0.10$, $p^{**} = 0.02$) | -0.042 | -0.030 | 0.000 | 0.026 | 0.035 |
| $p^{**} = 0.01$ | 2PLM-MLE | -0.028 | -0.039 | -0.034 | -0.057 | -0.061 |
| | 3PLM-MLE | -0.673 | -0.611 | -0.628 | -0.594 | -0.580 |
| | 4PLM-MLE | -0.635 | -0.542 | -0.522 | -0.446 | -0.425 |
| | 4PLM-Robust($p^{**} = 0.00$) | -0.087 | -0.082 | -0.078 | -0.077 | -0.072 |
| | 4PLM-Robust($p^{**} = 0.01$) | 0.012 | -0.031 | 0.005 | 0.023 | 0.005 |
| | 4PLM-Robust($p^{**} = 0.03$) | 0.054 | 0.033 | 0.024 | 0.056 | 0.089 |
| | BIW (B = 4) | -0.043 | -0.028 | -0.032 | -0.029 | -0.023 |
| | Huber (H = 1) | -0.034 | -0.021 | -0.050 | -0.042 | -0.036 |
| | 4PLM-Robust ($p^* = 0.10$, $p^{**} = 0.02$) | -0.086 | -0.049 | -0.040 | -0.011 | 0.019 |
| $p^{**} = 0.03$ | 2PLM-MLE | -0.059 | -0.079 | -0.082 | -0.110 | -0.128 |
| | 3PLM-MLE | -0.739 | -0.718 | -0.731 | -0.775 | -0.770 |
| | 4PLM-MLE | -0.664 | -0.616 | -0.618 | -0.585 | -0.543 |
| | 4PLM-Robust($p^{**} = 0.00$) | -0.062 | -0.112 | -0.118 | -0.126 | -0.132 |
| | 4PLM-Robust($p^{**} = 0.01$) | -0.016 | -0.012 | -0.010 | 0.010 | 0.009 |
| | 4PLM-Robust($p^{**} = 0.03$) | 0.030 | 0.041 | 0.047 | 0.049 | 0.043 |
| | BIW (B = 4) | -0.052 | -0.056 | -0.065 | -0.051 | -0.059 |
| | Huber (H = 1) | -0.077 | -0.069 | -0.078 | -0.062 | -0.081 |
| | 4PLM-Robust ($p^* = 0.10$, $p^{**} = 0.02$) | -0.139 | -0.115 | -0.119 | -0.105 | -0.084 |

*Note.* The biases are obtained by $(\sum \hat{\theta}_k)/5000 - \theta$ using 5,000 replications.

presents the biases of the 2PLM-MLE, the 4PLM-Robust, the BIW and the Huber estimations of the five fictive subjects.

The first section of Table 3 shows the results when $p^{**}$ is set at 0.00 for the response-generated model, indicating no occasional errors. The absolute values of the biases of the 2PLM-MLE, the BIW estimation, the Huber estimation, and the 4PLM-Robust ($p^{**} = 0.00$) of these subjects are quite small, close to zero. The absolute values of the biases of the 4PLM-Robust ($p^{**} = 0.01$) and the 4PLM-Robust ($p^{**} = 0.03$) of the five subjects are slightly larger. Additionally, the 3PLM-MLE and the 4PLM-MLE's biases are all negative values, whose absolute values are relatively larger, as observed in the other two sections of Table 3.

The second section of Table 3 shows the results when $p^{**} = 0.01$. The absolute values of the biases of the 4PLM-Robust ($p^{**} = 0.01$) are nearly zero, lower than the biases of the other methods. The absolute values of the biases of the 2PLM-MLE, the BIW and the Huber estimation are larger than those in the first section of Table 3, and the bias of the 2PLM-MLE is -0.061 at the high-ability level ($\theta = 2.000$).

The third section of Table 3 shows the results when $p^{**} = 0.03$. The absolute values of the biases of the 4PLM-Robust ($p^{**} = 0.03$) are nearly zero, lower than the biases of the other methods. All of the absolute values of the biases of the 2PLM-MLE, the 3PLM-MLE, the 4PLM-MLE, the 4PLM-Robust ($p^{**} = 0.00$), the 4PLM-Robust ($p^{**} = 0.01$), the BIW and the Huber estimations increase, compared with the second section of Table 3. The bias of the 2PLM-MLE is-0.128 at the high-ability level ($\theta = 2.000$), which is relatively larger.

In terms of each examinee's absolute values of biases in Table 3, the 4PLM-Robust ($p^* = 0.10$, $p^{**} = 0.02$) approximates the Huber and BIW estimation when $p^{**} = 0.00$. However, the 4PLM-Robust ($p^* = 0.10$, $p^{**} = 0.02$) is slightly larger than the Huber and BIW estimation when $p^{**} = 0.01$ and $p^{**} = 0.03$. That is, the 4PLM-Robust ($p^* = 0.10$, $p^{**} = 0.02$) is inferior to the Huber and BIW estimation. There are no random guessing responses in the simulation, but 4PLM-Robust ($p^* = 0.10$, $p^{**} = 0.02$) assumes that random guessing responses exist in the test, and the subjects are assumed to have guessed on the item if their expected probability of correct responding is below $p^*$ value at 0.10.

## Simulation Case III: Carelessness with simultaneous random guessing

In Simulation Case III, two types of response disturbances occur simultaneously, including guessing and carelessness. In fact, they often co-occur on practice tests.

Four combinations of guessing and carelessness in the response-generated model are discussed. Combination I is $p^* = 0.05$ and $p^{**} = 0.01$ in the generated model, signifying low guessing probability and low occasional error probability in the generated responses. Combination II is $p^* = 0.05$ and $p^{**} = 0.03$ in the generated model, indicating low guessing probability and high occasional error probability in the generated responses. Combination III is $p^* = 0.15$ and $p^{**} = 0.01$, indicating high guessing probability and low occasional error probability in the generated responses. Combination IV is $p^* = 0.15$ and $p^{**} = 0.03$, indicating high guessing probability and high occasional error probability in the generated responses.

In this study, the generated model of two types of response disturbances is presented as

$$P_{vi} = \begin{cases} 0.20 & \text{if } \psi(\theta_v - b_i) \leq p* \\ \psi(\theta_v - b_i) \bullet (1 - p**) & \text{if } \psi(\theta_v - b_i) > p* \end{cases}, \tag{17}$$

where $\psi(\theta_v - b_i)$ is the expected correct probability of subject $v$ using the 2PLM. The generated model is essentially the 2PLM plus the formula of guessing simultaneous with carelessness disturbance. In Eq (16), $p^*$ is set at 0.05 and 0.15, and $p^{**}$ is set at 0.01 and 0.03.

The approach using the 4PLM-Robust to robustify ability estimation is presented as

$$
\begin{cases}
c = \begin{cases} 0.20 & for \ \psi(\theta_v - b_i) \leq p* \\ 0 & \text{otherwise} \end{cases}, \\
\gamma = 1 - p** & \text{carelessness errors appear}
\end{cases}
\tag{18}
$$

where $p^*$ is set at 0.05 or 0.15 and $p^{**}$ is set at 0.01 or 0.03. Thus, the 4PLM-Robust have four forms, that is, 4PLM-Robust($p^* = 0.05$, $p^{**} = 0.01$), 4PLM-Robust($p^* = 0.05$, $p^{**} = 0.03$), 4PLM-Robust($p^* = 0.15$, $p^{**} = 0.01$), 4PLM-Robust($p^* = 0.15$, $p^{**} = 0.03$).

Table 4 shows the biases of Case III. All of the biases of the 2PLM-MLE in these four combinations at low ability ($\theta = -2$) are larger than 0.500, which is statistically significant. In Combination II, the bias of the 2PLM-MLE is -0.247 at high ability ($\theta = 2$), which signals a relatively large absolute value. In these four combinations, the absolute values of the biases of the BIW (B = 4) and the Huber (H = 1) are lower than those of the 2PLM-MLE. The biases of the 3PLM-MLE and the 4PLM-MLE are all negative values and below -0.350, and their absolute values are relatively larger.

In these four combinations, the absolute values of the biases of the BIW (B = 4) and the Huber (H = 1) are lower than those of the 2PLM-MLE. Furthermore, when the critical points of $p^*$ and $p^{**}$ in the 4PLM-Robust are equal to those in the response-generated model, the absolute values of the biases of the corresponding 4PLM-Robust are the smallest and close to zero among the four forms of the 4PLM-Robust, which also indicates that the 4PLM-Robust ($p^* = critical\ point$, $p^{**} = critical\ point$) is better than the other robust methods in the four combinations.

Especially for the 4PLM-Robust ($p^* = 0.10$, $p^{**} = 0.02$), $p^*$ and $p^{**}$ are also set to median value, that is, $p^* = 0.10$, $p^{**} = 0.02$. In terms of each examinee's absolute values of biases in Table 4, 4PLM-Robust ($p^* = 0.10$, $p^{**} = 0.02$) is less than the Huber estimation and is close to or slightly less than the BIW estimation.

## Discussion and conclusion

In the context of school training and teaching, students are generally told to randomly choose one of the alternatives if they do not know the correct answer or to choose the most likely answer when they are unsure. Hence, random guessing responses undoubtedly exist in practice tests, especially for low-ability and medium-ability subjects. Therefore, $p^*$ should be above zero, at 0.05, 0.10, or 0.15. Also, cheating, sharing answers on similar tests, and other reasons result in response disturbances similar to random guessing. Occasional error disturbance may result from many factors such as carelessness, transcription errors, and anxiety-exorbitant intention. Thus, the probability of occasional errors would probably be above zero on practice tests.

Over the last decades, some researchers have assumed that accounting for the 3PLM or the 4PLM by increasing model complexity could easily lead to large sample size requirements for estimation purposes (Hambleton & Swaminathan, 1985) [7]. In this paper, abilities are estimated using the 4PLM-Robust ($p^* = critical\ point$, $p^{**} = critical\ point$) in the three simulation studies when guessing and carelessness responses occur. If no guessing or carelessness responses exist, the 4PLM-Robust ($p^* = critical\ point$, $p^{**} = critical\ point$) can nevertheless be used because the 2PLM-MLE is a special case of the 4PLM-Robust with $p^* = 0$ and $p^{**} = 0$.

The two-parameter Logistic model, the three-parameter Logistic model, or the Rasch model can be considered reasonable approximations of the "true" unknown model. If response disturbances, including guessing or careless responses, actually exist, maximum-likelihood estimates of ability based on such models will yield biased ability estimates. For example, in

**Table 4. The biases in Case III: With simultaneous carelessness and random guessing.**

| Generated model | Method | True ability value | | | | |
|---|---|---|---|---|---|---|
| | | **-2.000** | **-1.000** | **0.000** | **1.000** | **2.000** |
| Combinations I | 2PLM-MLE | 0.675 | 0.465 | 0.263 | 0.053 | -0.086 |
| | 3PLM-MLE | -0.592 | -0.592 | -0.566 | -0.612 | -0.585 |
| | 4PLM-MLE | -0.512 | -0.498 | -0.450 | -0.450 | -0.426 |
| | **4PLM-Robust ($p^* = 0.05$, $p^{**} = 0.01$)** | **0.044** | **0.045** | **0.049** | **0.034** | **-0.004** |
| | 4PLM-Robust ($p^* = 0.05$, $p^{**} = 0.03$) | 0.174 | 0.200 | 0.142 | 0.107 | 0.099 |
| $p^* = 0.05$ | 4PLM-Robust ($p^* = 0.15$, $p^{**} = 0.01$) | -0.098 | -0.079 | -0.073 | -0.078 | -0.069 |
| | 4PLM-Robust ($p^* = 0.15$, $p^{**} = 0.03$) | -0.040 | -0.009 | -0.013 | -0.004 | 0.037 |
| | 4PLM-Robust ($p^* = 0.10$, $p^{**} = 0.02$) | -0.018 | -0.004 | 0.012 | 0.005 | 0.016 |
| $p^{**} = 0.01$ | BIW (B = 4) | 0.028 | 0.010 | 0.018 | 0.013 | -0.035 |
| | Huber (H = 1) | 0.183 | 0.111 | 0.074 | -0.019 | -0.038 |
| Combinations II | 2PLM-MLE | 0.630 | 0.389 | 0.136 | -0.096 | -0.247 |
| | 3PLM-MLE | -0.677 | -0.679 | -0.711 | -0.739 | -0.773 |
| | 4PLM-MLE | -0.621 | -0.574 | -0.563 | -0.566 | -0.549 |
| | 4PLM-Robust ($p^* = 0.05$, $p^{**} = 0.01$) | -0.023 | -0.018 | -0.088 | -0.091 | -0.093 |
| $p^* = 0.05$ | **4PLM-Robust ($p^* = 0.05$, $p^{**} = 0.03$)** | **0.021** | **-0.014** | **0.029** | **0.042** | **0.002** |
| | 4PLM-Robust ($p^* = 0.15$, $p^{**} = 0.01$) | -0.148 | -0.175 | -0.155 | -0.174 | -0.195 |
| $p^{**} = 0.03$ | 4PLM-Robust ($p^* = 0.15$, $p^{**} = 0.03$) | -0.062 | -0.051 | -0.091 | -0.122 | -0.083 |
| | 4PLM-Robust ($p^* = 0.10$, $p^{**} = 0.02$) | -0.070 | -0.088 | -0.085 | -0.074 | -0.076 |
| | BIW (B = 4) | -0.061 | -0.061 | -0.070 | -0.055 | -0.087 |
| | Huber (H = 1) | 0.118 | 0.052 | -0.019 | -0.077 | -0.120 |
| Combinations III | 2PLM-MLE | 0.772 | 0.563 | 0.393 | 0.176 | 0.014 |
| | 3PLM-MLE | -0.532 | -0.540 | -0.512 | -0.552 | -0.549 |
| | 4PLM-MLE | -0.439 | -0.417 | -0.387 | -0.370 | -0.364 |
| | 4PLM-Robust ($p^* = 0.05$, $p^{**} = 0.01$) | 0.274 | 0.221 | 0.175 | 0.138 | 0.083 |
| $p^* = 0.15$ | 4PLM-Robust ($p^* = 0.05$, $p^{**} = 0.03$) | 0.289 | 0.314 | 0.288 | 0.277 | 0.194 |
| | **4PLM-Robust ($p^* = 0.15$, $p^{**} = 0.01$)** | **-0.045** | **0.001** | **-0.003** | **-0.006** | **-0.012** |
| | 4PLM-Robust ($p^* = 0.15$, $p^{**} = 0.03$) | 0.072 | 0.079 | 0.077 | 0.084 | 0.075 |
| | 4PLM-Robust ($p^* = 0.10$, $p^{**} = 0.02$) | 0.091 | 0.098 | 0.104 | 0.100 | 0.089 |
| $p^{**} = 0.01$ | BIW (B = 4) | 0.105 | 0.084 | 0.102 | 0.086 | 0.024 |
| | Huber (H = 1) | 0.293 | 0.234 | 0.171 | 0.092 | 0.019 |
| Combinations IV | 2PLM-MLE | 0.721 | 0.512 | 0.328 | 0.050 | -0.136 |
| | 3PLM-MLE | -0.638 | -0.644 | -0.667 | -0.712 | -0.756 |
| | 4PLM-MLE | -0.549 | -0.523 | -0.515 | -0.517 | -0.503 |
| | 4PLM-Robust ($p^* = 0.05$, $p^{**} = 0.01$) | 0.159 | 0.157 | 0.080 | 0.059 | -0.030 |
| $p^* = 0.15$ | 4PLM-Robust ($p^* = 0.05$, $p^{**} = 0.03$) | 0.274 | 0.247 | 0.230 | 0.164 | 0.070 |
| | 4PLM-Robust ($p^* = 0.15$, $p^{**} = 0.01$) | -0.117 | -0.090 | -0.113 | -0.113 | -0.134 |
| | **4PLM-Robust ($p^* = 0.15$, $p^{**} = 0.03$)** | **-0.010** | **-0.024** | **-0.006** | **-0.005** | **-0.015** |
| | 4PLM-Robust ($p^* = 0.10$, $p^{**} = 0.02$) | 0.017 | 0.016 | 0.056 | 0.015 | 0.013 |
| $p^{**} = 0.03$ | BIW (B = 4) | 0.078 | 0.053 | 0.044 | 0.053 | -0.028 |
| | Huber (H = 1) | 0.221 | 0.178 | 0.092 | 0.026 | -0.057 |

*Note.* (1) The biases are obtained by $(\sum \hat{\theta}_k)/5000 - \theta$ using 5,000 replications.

(2) Bold type indicates the biases that are the least of all the estimations.

simulation study Case I, for the $p^*$ = 0.15 section, the biases using the 2PLM-MLE overestimate all of the subjects. However, in simulation study Case II, for the $p^{**}$ = 0.05 section, the biases using the 2PLM-MLE and the 3PLM-MLE underestimate all of the subjects.

Harberman (2006) observed that the guessing parameter in the 3PLM does not provide an evident gain compared with the 2PLM [19]. However, Harberman's study posited a guessing probability for each item, which was constant over all subjects. Because not all the subjects show their guesswork during tests, Mislevy and Bock (1982) illustrated that the 3PLM posited a guessing probability for each item, which was constant over all subjects but would be in error because it must under-reward those who do not guess at all [3]. The results of this paper validate Mislevy and Bock's (1982) conclusion. However, the subjects are not underestimated when using the 4PLM-Robust estimation rather than the 3PL-MLE or the 4PL-MLE. Thus, the 4PLM-Robust estimation, the BIW, the Huber estimation and other robust estimators are expected to play an important role in ability estimation when response disturbances really exist.

In this study, the example and three simulation studies suggest that the 4PLM-Robust estimation is an effective method for robust estimation of a latent trait in the presence of random guessing and carelessness. The advantages of the 4PLM-Robust estimation are as follows: (1) MLE and its Newton-Raphson method in the 4PLM-Robust estimation are identical to those in the 2PLM. The 4PLM-Robust estimation, by its nature, is an improvement of the 4PLM-MLE. (2) Because not all the subjects are believed to guess, the 4PLM-Robust estimation does not under-reward those who do not guess at all or over-reward heavy guessers. (3) Responses for guessing or carelessness need not be altered or deleted, unlike the BIW and the Huber estimations, which down-weight most of the responses. Thus, the 4PLM-Robust's calculation is simpler than the other two models' calculations.

For many years, it was believed that the $c$ and $\gamma$ parameters in the 4PLM model are complicated and hard to estimate. So far, the $c$ and $\gamma$ parameters are estimated by the Bayesian approach [8, 14]. So the critical-probability of the guessing parameter $p^*$ and the carelessness parameter $p^{**}$ on tests can possibly be computed in future studies by the Bayesian approach.

In fact, the tuning constant $B$ in the Biweight estimation can be set to 4.0, 2.0, 6.0, or the other constants, and the tuning constant $B$ for each item is possibly different. In fact, in the study of Mislevy and Bock (1982), they set the median value of $B$ to 4.0 for all the test items [3]. And the critical point constant $H$ in Biweight estimation can be set to 1.0, 2.0, 0.5, or the other tuning constants, and the constant $H$ for each item is possibly different. Nevertheless, in the study of Schuster and Yuan (2011), they set the median value of $H$ to 1.0 for all the test items. Similarly, in the study, a critical point of the probability of random guessing $p^*$ and the probability of occasional carelessness is also set to median value, that is, $p^*$ = 0.10, $p^{**}$ = 0.02. In the simulation case **III**, the 4PLM-Robust ($p^*$ = 0.10, $p^{**}$ = 0.02) is slightly superior to the Huber estimation and is close to or slightly superior to the BIW estimation in terms of the absolute values of biases for each examinee.

## Supporting information

**S1 Appendix.**
(DOC)

**S1 File.**
(RAR)

## Author Contributions

**Conceptualization:** Xiaozhu Jian, Dai Buyun.

**Data curation:** Xiaozhu Jian, Dai Buyun.

**Methodology:** Xiaozhu Jian, Dai Buyun, Deng Yuanping.

**Software:** Xiaozhu Jian.

**Validation:** Xiaozhu Jian.

**Visualization:** Deng Yuanping.

**Writing – original draft:** Xiaozhu Jian, Dai Buyun, Deng Yuanping.

**Writing – review & editing:** Xiaozhu Jian, Dai Buyun, Deng Yuanping.

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
