## [Decision Letter · Decision Letter 0]

23 Nov 2020

PONE-D-20-32376

The Robust Estimation of Examinee Ability Based on the Four-parameter Logistic Model When Guessing and Carelessness Responses Exist

PLOS ONE

Dear Dr. Jian,

Thank you for submitting your manuscript to PLOS ONE. After careful consideration, we feel that it has merit but does not fully meet PLOS ONE’s publication criteria as it currently stands. Therefore, we invite you to submit a revised version of the manuscript that addresses the points raised during the review process.

You should address all the comments of the reviewers carefully. Particularly, consult an expert to improve the language.

We look forward to receiving your revised manuscript.

Kind regards,

Lianmeng Jiao

Academic Editor

PLOS ONE

Journal Requirements:

Reviewers' comments:

Reviewer's Responses to Questions

**Comments to the Author**

1. Is the manuscript technically sound, and do the data support the conclusions?

Reviewer #1: No

Reviewer #2: No

Reviewer #3: Partly

2. Has the statistical analysis been performed appropriately and rigorously? 

Reviewer #1: No

Reviewer #2: No

Reviewer #3: Yes

3. Have the authors made all data underlying the findings in their manuscript fully available?

Reviewer #1: No

Reviewer #2: Yes

Reviewer #3: Yes

4. Is the manuscript presented in an intelligible fashion and written in standard English?

Reviewer #1: No

Reviewer #2: No

Reviewer #3: No

5. Review Comments to the Author

Reviewer #1: I have reviewed the manuscript 'The Robust Estimation of Examinee Ability Based on the Four-parameter Logistic Model When Guessing and Carelessness Responses Exist'. Although this manuscript addresses an interesting topic, I have detected (what for me are) several important limitations.

* I am not a native English speaker. I usually need the assistance of a professional translator with my own manuscripts. Having said that, I have found tha English level below what could and should be expected. For instance, I'm not sure about what this is supossed to mean: "According to their investigation, the Huber estimation, rather than the Biweight estimation, should be applied when reducing sampling variability was prior to reducing biases."

* The structure should be improved. For instance:

- "Waller (1974) and Wainer and Wright (1980) proposed approaches to the robust estimation of latent ability[1, 2], including traditional correction for guessing, Jackknife, AMT-Robustified Jackknife, and WIM. AMT-Robustified Jackknife was generally considered the best and most efficient method for tests comprising forty or fewer items[2]." The authors here list several alternatives (from rather old sources) that are never described. Why do the reader need this apparently irrelevant information? Why, if the AMT-Robustified Jackknife was considered the best alternative, is not used in the present study?

- Much of the details of the estimation approach described in the simulations studies should be placed in the Introduction, where those models are supposed to be described.

* In the section 'Recent literature on the 4PLM', the authors should clearly differentiate the results from computerized adaptive testing from those from non-adaptive applications. Up to now, those two lines of research are combined.

* Authors duplicate information. For instance, r = a(theta - b) can be found in page 4, 6, 9...

* Equations in page 7 cannot be properly read, as some elements have been moved.

* From my point of view, authors should not use BILOG to estimate trait levels. Or, at least, they should provide evidence that their Visual Basic routines, when programmed for 2PLM, BIW, and 3PLM, lead to the same results as those from BILOG. Otherwise, we don't know if the differences can be attributed to software differences or to the estimation methods in themselves.

* Authors should justify why they are using maximum-likelihood and not any of the Bayesian alternatives, as expected a posteriori with a uniform prior.

And, more importantly:

* I really don't know what can be learnt from the 'Example' study. If there is no correct theta level, what are the readers expected to learn from this? The authors used clearly incorrect models (that would have never been estimated if the correct model was 2PLM), as the 3PLM with c = .20. What is the point of doing that? Isn't obvious that using an incorrect model can distort trait level estimates?

* With respect to the simulation studies, it was difficult for me to correctly evaluate them, as the provided description is very difficult to follow. If the new method is correctly described in the Introduction, the description of the design of those studies could be greatly simplified.

Apparently, the authors ahve found that the parameters are better recovered when the model used to generate the data is also the model used to recover data. Perhaps I'm missing something relevant, but, again, this seems as obvious.

* I would greatly recommend this article:

Green, B. F. (2011). A comment on early student blunders on computer-based adaptive tests. Applied Psychological Measurement, 35(2), 165-174.

There some very pertinent points about this area of research were raised. I would not repeat them here.

Reviewer #2: The paper introduces robust estimation method for the four-parameter logisitic model (4PLM) to account for guess and careless responses. While I think I could understand the premise and purpose, I am afraid I became lost quite early either due to mistakes, or lack of clarity in derivations and reasoning. I think the paper would benefit from being much clearer about what is new here, and what others have done, but also please have a close look at formatting, potential errors in equations and maybe notations. Have some comments below that could perhaps help start this, and if I wrong with these then perhaps just more clarity is needed.

Main comments

1. Formatting of (5), (6) and (7) needs fixing. It is difficult to determine what the equations should be.

2. On line 117: why is P(ri) now exp(ri)/(1 - exp(ri))? Isn't is exp(ri)/(1 + exp(ri)) since it is a probability. Then line 119 onwards as well do maybe I am missing something here. With the formatting issues for (5), (6) and (7) I can't tell what is happening here, and whether this makes a difference.

3. I got lost with some of the notation, which had me second guessing at times. E.g. what is eta_i? E.g. in (5) isn't is just the residual ri?

4. In the two sections starting Line 105, it wasn't clear to me what was new to this paper, and what has previously been introduced by others. Is it just (8)?

Minor (example minor points)

The paper needs to some careful attention to grammatical errors and formatting issues. I have some examples below.

While 3PLM and 4PLM are defined in the abstract, they also need to be defined in the intro before use. 2PLM also needs to be defined.

Shouldn't rv on line 62 also be indexed by i? Later, ri used to indicate item i and an arbitrary subject.

line 106: "approaches" should "approach"

Eq 4: uj?

line 117: Stat of new sentence, "let" should be "Let" and need to end the sentence after Eq. 4.

Line 119: new sentence, "if" to "If"

Line 119 onwards: again exp(ri)/(1 - exp(ri)) so maybe I am missing something here

Reviewer #3: The manuscript addresses the problem of estimate the ability parameter in the presence of response disturbance, such as guessing, cheating, careless and transcription errors. The authors propose a new model that would be robust in the presence of that kind of occurrence and compare its results with the standard models present in Item Response Theory literature: 2PLM-MLE, 3PLM-MLE, the 4PLM-MLE, the Biweight estimation and the Huber estimation. It is worth mentioning that provide accurate ability estimation even in the presence of disturbance is a core matter in education assessment. The article is able to present the methodology and results quite clear. Therefore, I would recommend the publication if the following changes and some clarifications are met:

#Major changes ###########################################################

The major shortcoming of the manuscript is that the simulation study does not provide a fair comparison between the models. That is, it assumes that the items parameters are all known in advance but, for some models, it does not match with the true generated data. Indeed, the best results are obtained, in each case, as expected, by the model that matches with the true generated data. For example, is assumed in advance that de c parameter in the 3PL-MLE model follows a normal distribution N(0.2, 0.01). That is, the c parameter will almost surely be located between 0.17 and 0.23. Nevertheless, in simulation Case II the c parameter of the true generated model is always zero. In a real situation, the items parameters are obtained previously through a pre-test and c parameter could be near zero in 3PL-MLE. Indeed, the parameters of each item are supposed to represent the empirical reality. I would recommend that, for each Case, the authors generate two data sets. The first data set would be used only to estimate the items parameters for each model. Therefore, these items were pre-tested. Then, you can use the second data set to estimate the ability parameter using the items parameters previously obtained. This would reproduce what actually occur in practice. Anyway, this is only a suggestion; the important point is to provide a fair comparison, using this solution or another.

I also suggest to include a simulation (Case IV) in which the true generated model follow exactly the 3PL-MLE or the 4PL-MLE to see the behavior of the other models in that situation. Finally, I recommend to include a standard 4PLM-Robust in all simulation Case, for example 4PLM-Robust(p *=0.10, p** =0.02) to see its behavior, since the authors did not provide a way to estimate p* and p** and, in practice, the user would need to inform this values.

#Minor changes ##########################################################

Introduction: There is no mention about the proposal model in the introduction or why is necessary a new approach to deal with ability estimation in presence of response disturbance (guessing, carelessness, etc.). It’s not clear whether there are shortcomings in the current models used to address the problem.

Line 84: It’s quite strong to state that anyone can hinder the development of a statistical model for a quarter of a century.

The follow mathematical equations/symbols are misconfigured: 6 to 7 are (lines 119 – 125); equations 9 to 10 (lines 164 to 167); equation 17 (line 422); line 424 – 425, line 496 - 500.

Line 161: “r, and”?

Line 171: “4:0” = 4.0

Line 172: If B =0 instead of If r_i = 0?

Line 252 and Line 286: d = γ? Keep the same notation throughout the manuscript.

In Table 2 (line 325), Table 3 (Line 382), Table 4 (line 424) the bias is provided, I would recommend also include the root mean square error (RMSE).

Line 402: the value -0.128 is considered statistically significant, please provide further information about that statement. That is, did the authors perform any statistical test?

Line 433: The statement “are below zero, and their absolute values are larger than 0.100” is not correct, according to Table 4.

Line 437: The statement “and below -0.500” is not correct, according to Table 4.

Line 459: the word “certainly” may imply in a too strong statement.

6. PLOS authors have the option to publish the peer review history of their article (what does this mean?). If published, this will include your full peer review and any attached files.

Reviewer #1: No

Reviewer #2: No

Reviewer #3: No

---

## [Author Response · Author response to Decision Letter 0]

25 Jan 2021

Response to Reviewers

Review Comments to the Author

Reviewer #1:

 I have reviewed the manuscript 'The Robust Estimation of Examinee Ability Based on the Four-parameter Logistic Model When Guessing and Carelessness Responses Exist'. Although this manuscript addresses an interesting topic, I have detected (what for me are) several important limitations.

* I am not a native English speaker. I usually need the assistance of a professional translator with my own manuscripts. Having said that, I have found that English level below what could and should be expected. For instance, I'm not sure about what this is suppossed to mean: "According to their investigation, the Huber estimation, rather than the Biweight estimation, should be applied when reducing sampling variability was prior to reducing biases."

Response：" According to their investigation, the Huber estimation, rather than the Biweight estimation, should be applied when reducing sampling variability was prior to reducing biases." The sentence is the original views of Schuster and Yuan (2011).

* The structure should be improved. For instance:

- "Waller (1974) and Wainer and Wright (1980) proposed approaches to the robust estimation of latent ability[1, 2], including traditional correction for guessing, Jackknife, AMT-Robustified Jackknife, and WIM. AMT-Robustified Jackknife was generally considered the best and most efficient method for tests comprising forty or fewer items[2]." The authors here list several alternatives (from rather old sources) that are never described. Why do the reader need this apparently irrelevant information? Why, if the AMT-Robustified Jackknife was considered the best alternative, is not used in the present study?

Response：

(1) The author just expounded the history of the researches about several alternative robust estimations in the introduction. 

(2) The AMT robustified jackknife is just the better one of the several previous methods. AMT robustified jackknife is only suitable for short tests. Moreover, the AMT robustified jackknife method is more complex, and the meaning of the AMT robustified jackknife is not exact. After the Biweight estimation was proposed, researchers do not use the AMT robustified jackknife method.

* Much of the details of the estimation approach described in the simulation studies should be placed in the Introduction, where those models are supposed to be described.

Response： It is revised in the new manuscript. The details of the estimation are already placed in the Introduction.

* In the section 'Recent literature on the 4PLM', the authors should clearly differentiate the results from computerized adaptive testing from those from non-adaptive applications. Up to now, those two lines of research are combined.

Response：It is revised in the new manuscript.

* Authors duplicate information. For instance, r = a(theta - b) can be found in page 4, 6, 9...

Response：The two sentences are deleted in the new manuscript. Those are:

The sentence on page 4 is deleted : Let ; thus represents the distance between subject and item in the units of the standard deviation of item .

The sentence on page 6 is deleted: indicate the residual between estimated ability and item difficulty. 

* Equations in page 7 cannot be properly read, as some elements have been moved.

Response：It has been all revised in the new manuscript.

* From my point of view, authors should not use BILOG to estimate trait levels. Or, at least, they should provide evidence that their Visual Basic routines, when programmed for 2PLM, BIW, and 3PLM, lead to the same results as those from BILOG. Otherwise, we don't know if the differences can be attributed to software differences or to the estimation methods in themselves.

Response： When programmed for 2PLM, BIW, and 3PLM using the Visual Basic program, it is found that the ability estimation results using the Visual Basic program are the same as those using BILOG. 

Subject

# Responses Visual Basic Program /BILOG Visual Basic Program

 2PLM

-MLE BIW

B=4 3PLM

-MLE 4PLM

-MLE 4PLM

-Robust Huber

H=1

1 1111100000 0.061 0.048 -0.221 -0.133 0.068 0.052

2 1111110000 0.763 0.710 0.538 0.627 0.724 0.711

3 1111101000 0.763 0.746 0.436 0.494 0.762 0.711

4 1111100100 0.763 0.653 0.262 0.131 0.206 0.666

5 1111100010 0.763 0.421 0.061 -0.060 0.096 0.346

6 1111100001 0.763 0.163 -0.080 -0.114 0.073 0.257

7 1111000000 -0.755 -0.726 -0.999 -0.886 -0.709 -0.720

8 1110100000 -0.755 -0.798 -1.239 -1.352 -0.684 -0.720

9 1101100000 -0.755 -0.617 -1.437 -1.817 -0.479 -0.435

10 1011100000 -0.755 -0.160 -1.570 -0.535 -0.022 -0.224

11 0111100000 -0.755 0.048 -1.642 -0.189 0.056 -0.146

* Authors should justify why they are using maximum-likelihood and not any of the Bayesian alternatives, as expected a posteriori with a uniform prior.

Response： 

The equation of expected a posteriori estimation is shown as 

 (1)

And , , is called a "node." Each node has an associated weight that takes into account the height of the density function in the neighborhood of , and the width of the rectangles. The values of , and are found by solving a set of equations that involve the continuous distribution to be approximated and the specified number of nodes. 

 However, in Biweight estimations, the weight function is shown as 

 (2)

How did the add to the equation (1) is still to be discussed to confirm, thus the Biweight estimations using the expected a posteriori with a uniform prior is in doubt. so is the Bayesian alternatives using the expected a posteriori with a uniform prior.

So the ability is estimated only by using maximum-likelihood estimation in the manuscript.

And, more importantly:

* I really don't know what can be learnt from the 'Example' study. If there is no correct theta level, what are the readers expected to learn from this? The authors used clearly incorrect models (that would have never been estimated if the correct model was 2PLM), as the 3PLM with c = .20. What is the point of doing that? Isn't obvious that using an incorrect model can distort trait level estimates?

Response：(1)the 'Example' study in the manuscript is similar to the 'Example' in Table 1 in the Schuster and Yuan (2011), and is also similar to the 'Example' in Table 1 in the Mislevy and Bock (1982).

The authors designed that the response pattern of subject #1 is designated as the referenced pattern. His estimated ability is 0.061 using the 2PLM-MLE, which is perceived as the referenced ability value.

The item difficulty of the 10-th item ( ) is greatly smaller than the ability of Subject #6, but Subject #6 gives a correct response to the 10-th item by guessing probably and gets a same ability value as Subject #2. So there is a guessing response probably, and the ability estimation of Subject #6 is not robust.

The item difficulty of the 1-st item ( ) is greatly larger than the ability of Subject #11, but Subject #11 gives a wrong response to the 10-th item because of carelessness or other error probably and gets a same ability value as Subject #7. So there is a carelessness response probably, and the ability estimation of Subject #11 is not robust.

In summary, the 'Example' can show the guessing and carelessness responses, and the authors try to compare the several robust estimation methods.

Table 1. The Biases of the Ability Estimates of the Five Approaches

Subject

# Responses Visual Basic Program /BILOG Visual Basic Program

 2PLM

-MLE BIW

B=4 3PLM

-MLE 4PLM

-MLE 4PLM

-Robust Huber

H=1

1 1111100000 0.061 0.048 -0.221 -0.133 0.068 0.052

2 1111110000 0.763 0.710 0.538 0.627 0.724 0.711

3 1111101000 0.763 0.746 0.436 0.494 0.762 0.711

4 1111100100 0.763 0.653 0.262 0.131 0.206 0.666

5 1111100010 0.763 0.421 0.061 -0.060 0.096 0.346

6 1111100001 0.763 0.163 -0.080 -0.114 0.073 0.257

7 1111000000 -0.755 -0.726 -0.999 -0.886 -0.709 -0.720

8 1110100000 -0.755 -0.798 -1.239 -1.352 -0.684 -0.720

9 1101100000 -0.755 -0.617 -1.437 -1.817 -0.479 -0.435

10 1011100000 -0.755 -0.160 -1.570 -0.535 -0.022 -0.224

11 0111100000 -0.755 0.048 -1.642 -0.189 0.056 -0.146

Note. 3PLM-MLE indicates c=0.20 for items 1~10 using the 3PLM.

4PLM-MLE indicates c=0.20 , =0.95 for items 1~10 using the 4PLM. 

4PLM-Robust indicates 4PLM-Robust ( , ).

* With respect to the simulation studies, it was difficult for me to correctly evaluate them, as the provided description is very difficult to follow. If the new method is correctly described in the Introduction, the description of the design of those studies could be greatly simplified.

Apparently, the authors have found that the parameters are better recovered when the model used to generate the data is also the model used to recover data. Perhaps I'm missing something relevant, but, again, this seems as obvious.

Response：

Firstly, the model used to generate the data of the guessing and carelessness response disturbances is also used or acknowledged by the researchers in the past year. 

Secondly, the 4PLM-Robust estimation and several other robust estimation methods are compared by the same simulation data.

Thirdly, the form of the 4PLM-Robust estimation is only similar to the test data generated model, but not the same. The 4PLM-Robust estimation and the test data generated model originate in the same Production principle about guessing and carelessness response. So, the form of the 4PLM-Robust estimation is only similar to the test data generated model. 

* I would greatly recommend this article:

Green, B. F. (2011). A comment on early student blunders on computer-based adaptive tests. Applied Psychological Measurement, 35(2), 165-174.

There some very pertinent points about this area of research were raised. I would not repeat them here.

 Response：Green’s research is cited in the new manuscript. 

In Green’ views, there are rare errors on middle difficulty items for very proficient test-takers, and the so-called bias is one of selective data analysis(Green, 2011). 

The authors believe that whether the score penalty for very proficient test-takers is statistically significant, the faultiness of underestimating them in the Rasch model or 2PLM in the test has been found, and the researchers should admit it but not overlook it. Furthermore, the reasons and methods should also be found in different research angles.

Reviewer #2: 

The paper introduces robust estimation method for the four-parameter logisitic model (4PLM) to account for guess and careless responses. While I think I could understand the premise and purpose, I am afraid I became lost quite early either due to mistakes, or lack of clarity in derivations and reasoning. I think the paper would benefit from being much clearer about what is new here, and what others have done, but also please have a close look at formatting, potential errors in equations and maybe notations. Have some comments below that could perhaps help start this, and if I wrong with these then perhaps just more clarity is needed.

Main comments

1. Formatting of (5), (6) and (7) needs fixing. It is difficult to determine what the equations should be.

Response：The sequence numbers of the following mathematical equations are misconfigured, including Equation (4), (5), (6) and (7). They are all revised in the new manuscript.

2. On line 117: why is P(ri) now exp(ri)/(1 - exp(ri))? Isn't is exp(ri)/(1 + exp(ri)) since it is a probability. Then line 119 onwards as well do maybe I am missing something here. With the formatting issues for (5), (6) and (7) I can't tell what is happening here, and whether this makes a difference.

Response： “exp(ri)/(1 - exp(ri))” is a mistake in the original manuscript. It is revised in the new manuscript.

 Line 119 onwards，the authors aimed to explain that 4PLM-Robust estimation originates in 2PLM and can be simplified as 2PLM. In other words, 2PLM is one special case of 4PLM-Robust estimation.

 so in the new manuscript，it is revised as：

 , (8)

where and . 

If guessing and carelessness response disturbances don’t occur simultaneously on the -th item，then and . If and , then and in Equation 8. If and , 4PLM-Robust estimation becomes 4PLM. That is, 4PLM-Robust estimation originates in 2PLM and can be simplified as 2PLM. In other words, 2PLM is the special case of 4PLM-Robust estimation.

3. I got lost with some of the notation, which had me second guessing at times. E.g. what is eta_i? E.g. in (5) isn't is just the residual ri?

Response：They are all revised in the new manuscript. indicates the residual between estimated ability and item difficulty. in Equation (5), (6), (7) are deleted.

 (5)

 (6)

 (7)

4. In the two sections starting Line 105, it wasn't clear to me what was new to this paper, and what has previously been introduced by others. Is it just (8)?

Response：The authors aimed to explain that 4PLM-Robust estimation originates in 2PLM and can be simplified as 2PLM. That is, 2PLM is the special case of 4PLM-Robust estimation. 

 And more, the Biweight and Huber robust estimation are introduced in Line 163-178 in the original manuscript (Line 192-217 in the new manuscript). The Biweight and Huber robust estimation can be simplified as 2PLM according to Equation 5. In other words, 2PLM is also the special case of Biweight and Huber robust estimation. 

Minor (example minor points)

The paper needs to some careful attention to grammatical errors and formatting issues. I have some examples below.

While 3PLM and 4PLM are defined in the abstract, they also need to be defined in the intro before use. 2PLM also needs to be defined.

Response： It is revised in the abstract .

Shouldn't rv on line 62 also be indexed by i? Later, ri used to indicate item i and an arbitrary subject.

Response： is revised to 

line 106: "approaches" should "approach"

Response： It is revised.

Eq 4: uj?

Response： It is revised to .

 (4)

line 117: Stat of new sentence, "let" should be "Let" and need to end the sentence after Eq. 4. 

Response： It is revised.

Line 119: new sentence, "if" to "If"

Response： It is revised.

Line 119 onwards: again exp(ri)/(1 - exp(ri)) so maybe I am missing something here

Response：In Line 119 in original manuscript, is revised to 

Reviewer #3: 

The manuscript addresses the problem of estimate the ability parameter in the presence of response disturbance, such as guessing, cheating, careless and transcription errors. The authors propose a new model that would be robust in the presence of that kind of occurrence and compare its results with the standard models present in Item Response Theory literature: 2PLM-MLE, 3PLM-MLE, the 4PLM-MLE, the Biweight estimation and the Huber estimation. It is worth mentioning that provide accurate ability estimation even in the presence of disturbance is a core matter in education assessment. The article is able to present the methodology and results quite clear. Therefore, I would recommend the publication if the following changes and some clarifications are met:

#Major changes ###########################################################

The major shortcoming of the manuscript is that the simulation study does not provide a fair comparison between the models. That is, it assumes that the items parameters are all known in advance but, for some models, it does not match with the true generated data. Indeed, the best results are obtained, in each case, as expected, by the model that matches with the true generated data. For example, is assumed in advance that de c parameter in the 3PL-MLE model follows a normal distribution N(0.2, 0.01). That is, the c parameter will almost surely be located between 0.17 and 0.23. Nevertheless, in simulation Case II the c parameter of the true generated model is always zero. In a real situation, the items parameters are obtained previously through a pre-test and c parameter could be near zero in 3PL-MLE. Indeed, the parameters of each item are supposed to represent the empirical reality. I would recommend that, for each Case, the authors generate two data sets. The first data set would be used only to estimate the items parameters for each model. Therefore, these items were pre-tested. Then, you can use the second data set to estimate the ability parameter using the items parameters previously obtained. This would reproduce what actually occur in practice. Anyway, this is only a suggestion; the important point is to provide a fair comparison, using this solution or another.

I also suggest to include a simulation (Case IV) in which the true generated model follow exactly the 3PL-MLE or the 4PL-MLE to see the behavior of the other models in that situation. Finally, I recommend to include a standard 4PLM-Robust in all simulation Case, for example 4PLM-Robust(p *=0.10, p** =0.02) to see its behavior, since the authors did not provide a way to estimate p* and p** and, in practice, the user would need to inform this values.

 Response：(1) In the original manuscript, the item parameters are known and are not changed in the three simulation cases. The design of the original manuscript is similar tothat of Schuster and Yuan (2011) , and Mislevy and Bock (1982).

If the item parameters are changed for each case, then the item parameters in different test cases will likely influence examinees’ ability. Then, it will be more complicated for the simulation study. 

(2)If c parameter in the 3PL-MLE model follows a normal distribution N(0.2, 0.01) in the test simulation, thenthere will be another different simulation way of guessing responses generating model in the test. 

 Firstly, as we know, if the examinee knows the answer to the item or is sure to answer correctly (for example, the probability is greater than 25%), then the examinee will not guess the answer of the item randomly. It is be defined as Guessing Principle here.

Secondly, in the manuscript, random guessing is generated by the following model, Equation (12) (Mislevy & Bock, 1982):

 (12)

The generation model, Equation (12) tallies with the Guessing Principle.

In another generating model of guessing responses in 3PL-MLE model, for all subjects, including the high-ability subjects, the expected probability of correct responding will include the probability of randomly guessing, 0.20 or 0.25. So 3PL-MLE model does not tally with the Guessing Principle.

(3) if the true generated model follows exactly the 3PL-MLE or the 4PL-MLE in the test, the design and the aim of the manuscript will be complicated and blurry, and the first aim will becomea comparison between the two different generation models of guessing responses. Moreover, there is few common points between the two different generation models. 

(4) In Table 2 and Table 3，the simulation data about 4PLM-Robust(p *=0.10, p** =0.02) is added.

#Minor changes ##########################################################

Introduction: There is no mention about the proposal model in the introduction or why is necessary a new approach to deal with ability estimation in presence of response disturbance (guessing, carelessness, etc.). It’s not clear whether there are shortcomings in the current models used to address the problem.

 Response： The generated model about the carelessness error in the Schuster and Yuan (2011), , is accurate. For instance,for the low-ability examinees, if the response probability and , the probability will be , that is, . As is known , is paradoxical and illogical. The generated model about the carelessness error in the Schuster and Yuan (1982) only includes the high-ability examinees. 

In the article, the authors believe that the generated model about the carelessness error is proposed according to the Principle about Guessing and carelessness error, and so should the Robust estimation. 

Line 84: It’s quite strong to state that anyone can hinder the development of a statistical model for a quarter of a century.

Response：It has been revised. Perhaps, their comments influence the research. The research on the 4PLM stagnated for a quarter of a century.

The follow mathematical equations/symbols are misconfigured: 6 to 7 are (lines 119 – 125); equations 9 to 10 (lines 164 to 167); equation 17 (line 422); line 424 – 425, line 496 - 500.

Response： All has been revised.

Line 161: “r, and”?

Response：“ , and“ is deleted

Line 171: “4:0” = 4.0

Response：It is revised.

Line 172: If B =0 instead of If ri = 0?

Response： B is a divider in Equation 10, so B can’t be 0. 

Line 252 and Line 286: d = γ? Keep the same notation throughout the manuscript.

Response：It is revised.

In Table 2 (line 325), Table 3 (Line 382), Table 4 (line 424) the bias is provided, I would recommend also include the root mean square error (RMSE).

Response： There is a small difference between the root mean square error(RMSE) of the five examinees, and it is not statistically significant. 

Line 402: the value -0.128 is considered statistically significant, please provide further information about that statement. That is, did the authors perform any statistical test?

Response：The word “statistically significant“ is revised to “relatively larger”.

Line 433: The statement “are below zero, and their absolute values are larger than 0.100” is not correct, according to Table 4.

Response： The sentence “ Simultaneously, all the biases of the 2PLM-MLE at high ability ( =2) are below zero, and their absolute values are larger than 0.100. “ is deleted in the revised vision.

Line 437: The statement “and below -0.500” is not correct, according to Table 4.

Response：“and below -0.500” is revised to “and below -0.350” 

Line 459: the word “certainly” may imply in a too strong statement.

Response：“certainly“ is revised to “probably“

---

## [Decision Letter · Decision Letter 1]

16 Feb 2021

PONE-D-20-32376R1

The Robust Estimation of Examinee Ability Based on the Four-parameter Logistic Model When Guessing and Carelessness Responses Exist

PLOS ONE

Dear Dr. Jian,

Thank you for submitting your manuscript to PLOS ONE. After careful consideration, we feel that it has merit but does not fully meet PLOS ONE’s publication criteria as it currently stands. Therefore, we invite you to submit a revised version of the manuscript that addresses the points raised during the review process.

We look forward to receiving your revised manuscript.

Kind regards,

Lianmeng Jiao

Academic Editor

PLOS ONE

Additional Editor Comments (if provided):

The reviewers still have critical comments for your work. Please revise the paper by addressing all the comments.

Reviewers' comments:

Reviewer's Responses to Questions

**Comments to the Author**

1. If the authors have adequately addressed your comments raised in a previous round of review and you feel that this manuscript is now acceptable for publication, you may indicate that here to bypass the “Comments to the Author” section, enter your conflict of interest statement in the “Confidential to Editor” section, and submit your "Accept" recommendation.

Reviewer #3: (No Response)

2. Is the manuscript technically sound, and do the data support the conclusions?

Reviewer #3: No

3. Has the statistical analysis been performed appropriately and rigorously? 

Reviewer #3: No

4. Have the authors made all data underlying the findings in their manuscript fully available?

Reviewer #3: Yes

5. Is the manuscript presented in an intelligible fashion and written in standard English?

Reviewer #3: Yes

6. Review Comments to the Author

Reviewer #3: The manuscript was revised and improved in many aspects, however the major changes required was not properly addressed. The major shortcoming of the manuscript is that the simulation study does not provide a fair comparison between the models. Using 3PLM-MLE and 4PLM-MLE the parameters c and γ was considered known, but it doesn’t match with the true generated data. For example, in simulation case 2, γ=1-p** for the 4PLM-Robust, which matches the true generated data, but it was set to γ=0.97 for 4PLM-MLE, regardless of the true generated data. Since c and γ are parameters that should be estimated, doesn’t seem to be fair to assume a wrong value to them when it’s possible to inform the true value. In the previous example, γ should be 1-p** for 4PLM-MLE.

The manuscript is very interesting in the sense the present the 4PLM-Robust model, which seems to be more flexible to accommodate many patterns of response disturbance. However, the article does not contain simulations to verify that it is robust enough when the disturbance pattern does not follow its assumptions. For example, a simulation case in which the true generated model follow exactly the 3PL-MLE or the 4PL-MLE to see the behavior of the 4PLM-Robust. Therefore, the main conclusion is that the best model is the one that corresponds to the true generated-response model, as should be expected.

#----------------------------------------------------------------------------------------------------------------------------------

Minor changes:

Introdution – There is no mention to the proposed method.

Page 7 (lines 122-130) ,9 (line 180), 10 (lines 182, 201), 26 (line 446) – The equations are misconfigured.

Line 198: donot = do not

Line 247: Replace smaller for larger ?

Line 250: Replace larger for smaller?

Line 257: “The response of Subject #6 most seriously deviates from the referenced pattern of subject #1”. Is not the opposite?

7. PLOS authors have the option to publish the peer review history of their article (what does this mean?). If published, this will include your full peer review and any attached files.

Reviewer #3: No

---

## [Editor Report · Decision Letter 2]

5 Apr 2021

The Robust Estimation of Examinee Ability Based on the Four-parameter Logistic Model When Guessing and Carelessness Responses Exist

PONE-D-20-32376R2

Dear Dr. Buyun,

We’re pleased to inform you that your manuscript has been judged scientifically suitable for publication and will be formally accepted for publication once it meets all outstanding technical requirements.

Kind regards,

Lianmeng Jiao

Academic Editor

PLOS ONE
---

## [Editor Report · Acceptance letter]

12 Apr 2021

PONE-D-20-32376R2 

The Robust Estimation of Examinee Ability Based on the Four-parameter Logistic Model When Guessing and Carelessness Responses Exist 

Dear Dr. Buyun:

I'm pleased to inform you that your manuscript has been deemed suitable for publication in PLOS ONE. Congratulations! Your manuscript is now with our production department. 

Kind regards, 

on behalf of

Dr. Lianmeng Jiao 

Academic Editor

PLOS ONE